# Gain-of-Function p53N236S Mutation Drives the Bypassing of HRas^V12^-Induced Cellular Senescence via PGC–1α

**DOI:** 10.3390/ijms24043790

**Published:** 2023-02-14

**Authors:** Hao Yang, Ke Zhang, Yusheng Guo, Xin Guo, Kailong Hou, Jing Hou, Ying Luo, Jing Liu, Shuting Jia

**Affiliations:** Laboratory of Molecular Genetics of Aging and Tumor, Medical School, Kunming University of Science and Technology, 727 Jing Ming Nan Road, Kunming 650500, China

**Keywords:** p53 mutant, oncogene-induced senescence, PGC–1α

## Abstract

One of the key steps in tumorigenic transformation is immortalization in which cells bypass cancer-initiating barriers such as senescence. Senescence can be triggered by either telomere erosion or oncogenic stress (oncogene-induced senescence, OIS) and undergo p53- or Rb-dependent cell cycle arrest. The tumor suppressor p53 is mutated in 50% of human cancers. In this study, we generated p53N236S (p53S) mutant knock-in mice and observed that p53S heterozygous mouse embryonic fibroblasts (*p53^S/+^*) escaped HRas^V12^-induced senescence after subculture in vitro and formed tumors after subcutaneous injection into severe combined immune deficiency (SCID) mice. We found that p53S increased the level and nuclear translocation of PGC–1α in late-stage *p53^S/+^*+Ras cells (LS cells, which bypassed the OIS). The increase in PGC–1α promoted the biosynthesis and function of mitochondria in LS cells by inhibiting senescence-associated reactive oxygen species (ROS) and ROS-induced autophagy. In addition, p53S regulated the interaction between PGC–1α and PPARγ and promoted lipid synthesis, which may indicate an auxiliary pathway for facilitating cell escape from aging. Our results illuminate the mechanisms underlying p53S mutant-regulated senescence bypass and demonstrate the role played by PGC–1α in this process.

## 1. Introduction

Cellular senescence is permanent cell cycle arrest characterized by active metabolism and sustained cell viability. Senescence can be triggered by intrinsic and/or extrinsic stimuli, including telomere shortening (replicative senescence) and oncogene activation (OIS) [1,2]. OIS is thought to pose a cancer-initiating barrier in vivo by preventing cells with oncogenic mutations from progressing to a malignant state. Although OIS is a defense mechanism against tumorigenesis, many abnormal cells form tumors by escaping senescence. Several mechanisms that may contribute to bypassed senescence and tumorigenic transformation have been proposed, including loss of key tumor suppressor genes, such as p53 or RB, reactivation of telomere maintenance mechanisms, genomic instability, and mitochondrial and epigenetic changes.

The tumor suppressor p53 (encoded by *TP53* in humans) plays a significant role in many antitumor pathways, such as cell cycle arrest, apoptosis, senescence, and DNA repair [3,4]. p53 is mutated in approximately 50% of human cancers (IARC *TP53* database, version R20, July 2019) [5]. In general, mutant p53 might exert three types of phenotypic effects [6,7,8]. First, when both p53 alleles are mutated or the wild-type allele is lost (known as a loss of heterozygosity, LOH, in which the wild-type allele is either deleted or mutated), the tumor suppressor function of p53 is partially or completely abrogated, causing loss of function (LOF). Second, many mutant p53 isoforms inhibit the function of coexpressed wild-type p53. This dominant-negative (DN) effect is largely realized by the formation of tetramers comprising a combination of mutant and wild-type proteins with defective DNA binding or transactivation. Third, some mutant p53 proteins show a gain of function that is not an action of the wild-type p53 protein. These gain-of-function (GOF) properties actively contribute to various aspects of tumorigenesis.

p53 is inactivated mainly by missense mutations in the DNA-binding domain [6,9,10], resulting in the formation of a full-length protein with an altered conformation and attenuated capacity for sequence-specific binding to DNA [8,11]. The majority of missense mutations in human cancer are located in 4 hotspot regions at codons 129–146, 171–179, 234–260, and 270–287 [12,13]. In our previous work, a rarely studied missense mutation p53^N236S^ in mice (p53^N239S^ in humans, p53S) in a hotspot region was found to be highly expressed in three independent alternative lengthening of telomere (ALT) tumorigenic cell lines derived from senesced Werner syndrome mouse embryonic fibroblasts (MEFs) [14,15]. This finding suggested that p53S may be critical for cell escape from senescence and the progression of immortalization.

However, the mechanism by which p53S regulates senescence bypass and tumorigenesis remains unclear. A few studies have reported on GOF mechanisms by which mutant p53 regulates the senescence escape pathway. One study suggested that both p53^R172H^ and p53-null cells escape KRAS^G12^-induced growth arrest/senescence in pancreatic cancer; nevertheless, only mice expressing mutant p53^R172H^ developed pancreatic ductal adenocarcinoma (PDAC), which indicated a novel role played by mutant p53 that is different from that of p53 knockout [16]. Furthermore, a subsequent study showed that in contrast to inducing senescence, the KRAS-RAF-MEK-MAPK effector pathway activated CREB1, allowing physical interactions with mutant p53 to promote WNT/β-catenin signaling by upregulating FOXA1, leading to PDAC metastasis [17]. In a recent study, a partial loss-of-function p53 mutant, p53^E177R^, provided an early temporary defense against oncogenic KRAS during tumorigenesis but was insufficient to block malignant progression to advanced tumor stages [18].

In this study, by generating p53S knock-in mice, we observed that p53S heterozygous MEFs (*p53^S/+^*) escaped HRas^V12^-induced senescence. We further demonstrate that p53S upregulated the expression of PGC–1α (also known as PPARGC1A, a PPARG coactivator) and enhanced the interaction between PGC–1α and peroxisome proliferator-activated receptor γ (PPARγ) to increase the number and enhance the function of mitochondria, resulting in increased lipid metabolism, thus inducing cells to escape from cellular senescence and leading to tumor formation. Therefore, this study provides novel insights into the mechanisms through which p53 mutant GOF mediates senescence bypass and tumor progression.

## 2. Results

### 2.1. The p53^S/+^ MEFs Escape HRas^V12^-Induced Senescence and Develop LOH

*p53* mutation is usually accompanied by LOH during tumor progression [19], which suggests that the remaining wild-type allele might be inactivated by a specific type of stress, such as that induced by an oncogenic stimulus. We first transiently transfected *p53^S/+^* and *p53^+/+^* MEFs (with wild-type MEFs as controls) with an HRas^V12^ expression plasmid. As expected, oncogenic HRas^V12^ induced cell senescence in the *p53^+/+^* MEFs (Figure 1A) [1]. Interestingly, *p53^S/+^*+Ras MEFs showed senescent characteristics similar to those of *p53^+/+^*+Ras MEFs, such as a typical flattened cell morphology, slow proliferation, and increased SA-β-Gal activity (Figure 1A,B, 80.43% senescent cells among the wild-type controls vs. 68.75% senescent cells among early stage *p53^S/+^*+Ras cells). However, only a small number of homozygous *p53^S/S^*+Ras cells exhibited a senescent phenotype (Figure 1B, 2.97% senescent cells among the early-stage *p53^S/S^*+Ras cells). When transfected with HRas^V12^, the expression of p21 in both the *p53^S/+^* and *p53^+/+^* MEFs was increased (Figure 1C, p21). However, in the *p53^S/S^* MEFs, the p21 level was not increased by HRas^V12^ overexpression, which was ascribed to the loss of function of p53S, as we previously reported [14]. These data suggested that, similar to that in wild-type MEFs, the wild-type allele in *p53^S/+^* MEFs induced premature senescence to prevent tumorigenesis.

However, the percentage of senescent cells in *p53^S/+^*+Ras was markedly decreased after continuous serial passages in vitro for more than 1 month (approximately 10 passages) (Figure 1A,B, late-stage *p53^S/+^*+Ras cells), which suggested that the wild-type p53 allele may have failed to induce cell senescence in the late-stage cells. We evaluated the genotype of the late-stage *p53^S/+^*+Ras cells and found that the amplification of the wild-type allele had notably decreased (Figure 1D, *p53^S/+^*+Ras p10), which indicated that the WT *p53* alleles in *p53^S/+^*+Ras cells may have been deleted or mutated. To confirm the genotype of the late-stage cells, the relative copy number was determined via real-time PCR, and the results showed that the *p53S* allele in the late-stage *p53^S/+^*+Ras cells was twofold that in the *p53^S/+^* MEFs (Figure 1E), suggesting a *p53^S/S^* genotype not a *p53^S/−^* genotype in the late-stage *p53^S/+^*+Ras cells. These results indicated that the p53 wild-type allele may have been deleted via an LOH mechanism. We also performed a karyotyping experiment to detect genomic stability, and the results showed a high number of multiple chromosome aberrations, including aneuploidy (2N > 40), chromosome fusions, and double-strand breaks in late-stage *p53^S/+^*+Ras cells than in *p53^S/+^*, *p53^S/S^*, or *p53^S/S^*+Ras cells. The genomic instability of the *p53^S/+^*+Ras cells suggested a possible explanation for the LOH of the WT *p53* alleles.

Our previous work suggested that *p53^−/−^*+S+Ras cells (both p53S and HRas^V12^ were introduced into *p53^−/−^*MEFs) formed tumors by subcutaneous injection into SCID mice [14]. As expected, similar to the *p53^S/S^*+Ras cells, late-stage *p53^S/+^*+Ras cells formed tumors in SCID mice (Figure 1F) and the differences in tumor size were not significant (Figure 1G, weight *p* = 0.422, diameter *p* = 0.177). Together, these results suggest that under stress caused by oncogene activation, heterozygous *p53^S/+^* cells selectively delete the wild-type allele and thus increase their potential for tumor transformation.

### 2.2. p53S Upregulates PGC–1α and Its Nuclear Translocation in Late-Stage p53^S/+^+Ras Cells

To further understand the OIS bypass resulting from p53S mutation, we performed a global analysis of p53S-regulated genes by chromatin immunoprecipitation-on-chip (ChIP-on-Chip) and microarray technologies. A total of 162 common genes were identified in both the ChIP-on-Chip and microarray datasets. We mapped the protein interaction network with the central points set to *TP53* and found that PPARγ and PGC–1α may be key genes regulated by p53S (Appendix A). To verify whether PGC–1α is regulated by p53S and how this increase in expression is regulated, we compared the expression of PGC–1α in early-stage (ES) senescent cells, late-stage (LS) tumorigenic cells, and SCID tumor cells (obtained from SCID tumors formed by the subcutaneous injection of late-stage cells, Figure 1F). Our results indicated that the expression of PGC–1α increased in both LS cells and SCID tumor cells (Figure 2A,B) compared with that in ES cells. We also observed an increase in the level of the p53 protein in the LS cells, possibly due to the activation of the p53 mutant allele acquired after LOH (Figure 2A). Next, we asked whether p53S directly regulates the transcription of PGC–1α. ChIP and luciferase reporter vector experiments were performed, and the results demonstrated that p53S but not the WT p53 protein bound the PGC–1α promoter region (Figure 2C,D, respectively). We further analyzed the intracellular localization of PGC–1α by immunofluorescence staining (IF), and the results showed that the expression of PGC–1α was significantly higher and mostly concentrated in the nucleus in LS cells when compared with ES cells. This result was confirmed by nucleoplasmic separation experiments (Figure 2E–G). Together, these data suggested that p53S upregulated PGC–1α expression and promoted its nuclear translocation in *p53^S/+^*+Ras tumorigenic cells by directly binding to the PGC–1α promoter.

### 2.3. p53S Improves Mitochondrial Quality and Quantity in Late-Stage p53^S/+^+Ras Cells via PGC–1α

It was reported that PGC–1α, a transcriptional coactivator, plays an important role in mitochondrial synthesis [20]. Therefore, we questioned whether mitochondrial function differs between ES and LS cells. MitoTracker™ Green was utilized to count the number of mitochondria, and the results showed an increase in the number of mitochondria in LS cells compared with ES cells (Figure 3A, *p* < 0.01). Consistently, the mRNA levels of mitochondrion-specific genes, including *COX1*, *CYTB*, and *ND1,* was increased in LS cells (Figure 3B, *p* < 0.01). ATP production was also significantly increased in LS cells (Figure 3C, *p* < 0.01). Taken together, these data suggested that mitochondrial function was impaired in ES senescent *p53^S/+^*+Ras cells; however, in LS cells, the quantity and function of mitochondria were increased, which may have been associated with the upregulation of PGC–1α in late-stage cells.

To verify whether PGC–1α plays a key role in promoting mitochondrial function and mediating *p53^S/+^* cell escape from HRas^V12^-induced senescence, we used CRISPR/Cas9-based editing to genetically eliminate *PGC–1α* in LS cells (Figure 3D). By colony screening, we obtained two clones (F10 Target2 and C11 Target4) in which the *PGC–1α* gene was knocked down because of a frameshift mutation (Figure 3E, and Appendix A). Both *PGC–1α* downregulated clones grew at rates similar to that of the ES cells but at a much slower rate than the LS cells (Figure 3E). This reduced growth rate was likely due to a reduction in mitochondria number (Figure 3A, *p* < 0.01) and ATP production (Figure 3C, no significant difference) in *PGC–1α*-downregulated LS cells. In summary, PGC–1α upregulated the biosynthesis and function of mitochondria in LS cells, and these effects were closely associated with the escape of *p53^S/+^*+Ras cells from OIS.

### 2.4. PGC–1α Downregulates Autophagy Levels in Late-Stage Cells by Reducing ROS Levels

We explored the mechanism by which mitochondrial differences are induced in early- and late-stage *p53^S/+^*+Ras cells. We first examined the expression of genes downstream of PGC–1α that are associated with mitochondrial proliferation, such as *Nrf*–*1* and *Nrf*–*2*. However, the expression of *Nrf*–*1* and *Nrf*–*2* was found to be only slightly elevated in LS cells (Appendix A, *p* > 0.05), which suggested that other factors contributed to the enhanced mitochondrial number and function in LS cells. ROS, which are mainly generated during mitochondrial oxidative metabolism, are possible candidates for this action because (1) the increase in ROS levels is an important OIS mechanism, leading to DNA damage, cell cycle arrest, and cell autophagy [21], and (2) ROS can disrupt the mitochondrial membrane potential, leading to mitochondrial dysfunction [22]. Hence, we hypothesize that PGC–1α, a key modulator of mitochondrial ROS production, may enhance mitochondrial quality and quantity in late-stage *p53^S/+^*+Ras cells by repressing ROS production and ROS-induced autophagy.

To assess this possibility, we first examined ROS levels in ES and LS cells by DCFH-DA staining. The ROS level was significantly increased in ES cells compared with *p53^S/+^* control cells, suggesting that oncogene expression induced ROS production and led to cell senescence. However, ROS level was diminished to basal level in LS cells (Figure 4A, *p* < 0.01). PGC–1α has been reported to regulate the expression of various antioxidant enzymes to reduce ROS levels. The expression of the antioxidant enzymes Sod1, Sod2, Ant, and Ucp2 was significantly increased in LS cells compared with EL cells (Figure 4B, *p* < 0.05). Excessive ROS can disrupt the mitochondrial membrane potential, which eventually leads to mitochondrial dysfunction. We next compared the mitochondrial membrane potential. The mitochondrial membrane potential was decreased significantly in the ES cells, while that in the LS cells was maintained at a level similar to that of the *p53^S/+^* MEFs (Figure 4C, *p* < 0.01), suggesting that the mitochondrial membrane was restored in the LS cells.

Downregulated ROS-induced autophagy in LS cells may also explain OIS bypass. To assess this possibility, we first examined the levels of autophagy-related proteins. The levels of Atg−7 and Atg−12 and the ratio of LC3−II/LC3−1 in ES cells was increased compared with those in LS cells (Figure 4D). These data indicated lower autophagy activation in LS cells. To determine whether the autophagy–lysosome flux was disrupted in LS cells, we counted the number of lysosomes after LysoTracker™ Red staining. The number of lysosomes was significantly decreased in LS cells (Figure 4E, *p* < 0.001). Moreover, autophagy activation in *PGC–1α*-depleted F10 cells was comparable to that in ES cells (Figure 4D,E), suggesting a critical role for PGC–1α in inhibiting ROS-induced autophagy.

To confirm the role played by ROS in inducing autophagy and senescence in *p53^S/+^*+Ras cells, hydrogen peroxide (H_2_O_2_) treatment was used to induce persistently elevated ROS levels (Figure 4F). Autophagy and cell cycle arrest were induced when ROS levels were elevated by H_2_O_2_ treatment in LS cells (Figure 4G,H), suggesting that an increase in ROS levels may be critical to OIS_._ Taken together, the results indicate that the autophagy pathway was impaired in LS cells, which was related to reduced ROS levels caused by the activation of PGC–1α, leading to an increase in mitochondrial quality and quantity and promotion of *p53^S/+^*+Ras cell bypass of OIS.

### 2.5. p53S Regulates the Interaction between PGC–1α and PPARγ

PPARγ is a ligand-dependent nuclear transcription factor. Recent studies have shown that it plays a significant role in promoting tumorigenesis. PPARγ promoted metastatic prostate cancer through activation of lipid signaling pathways [23]. Since PGC–1α is a coactivator of PPARγ and is also the key gene regulated by p53S (Appendix A), we asked, does PPARγ also play a role in *p53^S/+^*+Ras cell bypassing senescence? We first monitored the expression of PPARγ and found that, similar to that of PGC–1α, PPARγ expression was high (Figure 5A) in LS cells and reduced in PGC–1α-knockdown cells (Figure 3G,H). In addition, PPARγ was also found to have highly accumulated in the nucleus of the LS cells. These results suggested that PPARγ and PGC–1α were synergistically involved in the senescence escape pathway of *p53^S/+^*+Ras cells. However, p53S did not directly regulate PPARγ expression by binding to its promoter region (Appendix A). To address the potential ability of p53S to regulate the interaction between PGC–1α and PPARγ, a co-IP assay was performed with 293T cells transfected with HA-tagged PGC–1α, FLAG-tagged PPARγ, and either MYC-tagged p53S or HIS-tagged WT p53. Our results suggested that overexpression of p53S, but not that of WT p53, enhanced the binding of PGC–1α and PPARγ in vitro (Figure 5B). To further examine whether the increased PPARγ activated lipid synthesis, Oil Red O staining was performed with ES and LS cells. The lipid content was significantly higher in the LS cells than in the ES *p53^S/+^*+Ras cells (Figure 5C,D, *p* < 0.05), suggesting that p53S regulated the senescence escape of the *p53^S/+^*+Ras cells by enhancing the binding of PGC–1α and PPARγ and promoting lipid synthesis. This metabolic change may affect the development and growth of tumors.

## 3. Discussion

OIS is a potent tumor-suppressive mechanism that halts the accelerated proliferation of cells with the potential for tumorigenic transformation. However, once the senescence barrier is overcome, oncogene expression drives cell transformation and tumor malignancy. The signaling pathways and molecular mechanisms that regulate these processes remain poorly understood and are important to cancer biology. The first observation of OIS was made through an in vitro study in which HRas^V12^ expression induced cellular senescence in a p53- or p16-dependent manner [1]. Subsequent work showed that loss of p53 permitted senescence bypass of cells in response to oncogene expression and greatly facilitated progression to malignant cancer [24]. However, very few studies have suggested that mutant p53 regulates the OIS bypass by cells. We observed that p53S lost transcriptional regulatory effects on both cell cycle arrest and apoptotic pathways and gained new functions that promoted tumorigenesis in vivo in cooperation with oncogenic Ras [14]. Most tumor suppressor genes undergo biallelic inactivation via deletion or truncation mutation during carcinogenesis, but *p53* is frequently (74%) inactivated by a single monoallelic missense mutation [8]. By generating a p53S heterozygous MEF cell (*p53^S/+^*), we observed that the overexpression of the HRas^V12^ oncogenic signal caused wild-type allele LOH and resulted in the escape from Ras-induced senescence and activation of tumor progression.

Moreover, at later stages of tumor development [19,25], when one p53 is mutated, the remaining wild-type allele is usually deleted (LOH) [26]. In addition, LOH upon wild-type p53 allele loss has been widely documented in many tumor spectra [27,28,29,30,31]. LOH has been observed in both contact mutant p53^R270H^ and the structural mutant p53^R172H^ cell lines to promote tumors [32]. LOH has been observed to be evident in fewer than one-half (44%) of the 36 tumors formed in 28 Li-Fraumeni syndrome (LFS) or Li-Fraumeni-like (LFL) patients who harbor germline mutations of the *p53* gene (two prominent mutant codons encode R273H and R175H in humans) [28]. In an LFS mouse model, LOH was observed in 10 of 19 tumors analyzed (4/10 *p53^R270H/+^* and 6/9 *p53^R172H/+^*) [33]. Similar to that in LFS mice, LOH was observed in 2 of 16 tumors of *p53^S/+^* mice. These reports suggested that, in addition to the LOH of missense mutant p53S, the acquisition of other mechanisms, such as GOF or DN, might be pertinent to the carcinogenic process. However, the specific context in which each mechanism is triggered is still unclear. Our study suggests that the overexpression of oncogenes might be important to LOH initiation (Figure 6).

ROS play critical roles in OIS. RasV12 expression results in senescence by upregulating mitochondrial ROS, which has been shown to be partially inhibited when ROS production is decreased [34]. In addition, overexpression of the BRAF^V600E^ oncogene caused pyruvate dehydrogenase (PDH) activation, which enhanced the use of pyruvate in the tricarboxylic acid cycle and increased the generation of oxidants, leading to OIS [35]. Several studies demonstrated that mitochondrial dysfunction induced cellular senescence [36]. This outcome was likely due to the increased production of oxidants through mitochondrial metabolism alterations. In the present study, we provide evidence that in late-stage *p53^S/+^*+Ras cells, ROS and ROS-related autophagy was repressed, and mitochondrial quality and quantity were greatly enhanced compared with those in early-stage senescent cells (Figure 6). Our findings support the idea that ROS levels are switches that regulate OIS escape.

Our findings also shed light on the role played by PGC–1α and PPARγ in modulating the OIS escape process. DePinho et al. showed that p53, which is induced by dysfunctional telomeres, bound to the promoters of PGC–1α and PGC−1β and repressed their expression, leading to mitochondrial compromise and cardiac aging [37,38]. In addition, PPARβ/δ, PPAR isoforms, promoted senescence and suppressed carcinogenesis by repressing HRas-induced ER (endoplasmic reticulum) stress [39]. Moreover, we identified PGC–1α as a factor involved in p53S-regulated OIS bypass of *p53^S/+^*+Ras cells, and suggested that this effect may be driven by reductions in ROS levels and autophagy activation that enhance mitochondrial quality and quantity. In addition, we demonstrated in another work that p53S might initiate mitophagy to clear up damaged mitochondria in response to hypoxic stress [40], which further supports our conclusion that p53S can promote cell proliferation in the presence of stress by improving the state of mitochondria (Figure 6). We also showed that p53S regulated the interaction between PGC–1α and PPARγ, providing further evidence that the p53S gain of function facilitates cell senescence bypass. We propose that this p53S–PGC–1α/PPARγ axis contributes to metabolic reprogramming and diminishes the damage caused by OIS.

## 4. Materials and Methods

### 4.1. Mice and Tumor Cells Harvest

SCID mice were purchased from the National Resource Center of Model Mice (NRCMM, Nanjing, China). A total of 1 × 10^6^ cells were injected subcutaneously into each site of SCID mice. When the size of the largest tumor reached 1 cm^3^, the mice were sacrificed, and the tumors were collected and digested in a tumor-digesting cocktail (4 mg/mL collagenase D and 4 mg/mL dispase II). Isolated tumor cells were plated and cultured in DMEM supplemented with 10% fetal bovine serum.

All mouse procedures were performed with the approval of the Animal Care and Use Committee of the Kunming University of Science & Technology (approval ID: M2018-0008).

### 4.2. Cell Lines and Constructs

All cell lines were cultured in DMEM supplemented with 10% fetal bovine serum (HyClone, Logan, UT, USA) in an incubator with 3% oxygen and 5% CO_2_ at 37 °C. A total of 200 μM H_2_O_2_ was used to treat the indicated cells for 5 days.

The pBabe-HRas^V12^ construct (as described previously [1]) was introduced into *p53^+/+^*, *p53^S/+^,* and *p53^S/S^* MEFs. We collected cells after 24 h, 48 h, 96 h, 8 days, 11 days, 18 days, and after more than 1 month of antibiotic selection.

The promoter region in PGC–1α (−2533~+78) from C57/BL6 mouse genomic DNA was amplified with the following oligos: BgLII-PGC–1α-F: 5′-GGAAGATCTTCCCGGTACCCCTGTGCTCTCTCTAGCTTCACA-3′ and EcoRI-PGC–1α-R: C5′-CGGAATTCCGGGCTCGAGCCAGCTCCCGAATGACGCCAGTC-3′. The PCR product was digested with BgLII/EcoRI and cloned into a pEZX-GA01 reporter vector.

The retrovirus vector pQXCIP (Clontech, San Jose, CA, USA) was used to induce PGC–1α and PPARγ protein production in 293T cells.

### 4.3. Western Blot Analysis

Cells were lysed in RIPA buffer containing a protease inhibitor cocktail kit (Roche, Basel, Switzerland), and the protein concentration was determined using a BCA kit (Thermo Fisher Scientific, Waltham, MA, USA). Twenty micrograms of total protein was separated by SDS–PAGE and then transferred to a PVDF membrane. After blocking in 10% skim milk for 1 h at room temperature, the membrane was incubated with primary antibodies overnight at 4 °C or 2 h at room temperature. The membrane was then incubated with horseradish peroxidase-labeled secondary antibodies and visualized with ECL. The cytoplasmic and nuclear protein fractions were extracted using a nuclear and cytoplasmic protein extraction kit (Beyotime Biotechnology, Shanghai, China) according to the manufacturer’s instructions.

The antibodies used for Western blotting were anti-Ras (1:500, BD Transduction Laboratories, San Jose, CA, USA), anti-p21 (1:500, Santa Cruz Biotechnology, Paso Robles, CA, USA), anti-p53 (1:250, Labvision Neomarker, Fremont, CA, USA), anti-phospho-p53 (Ser15) (1:500, Cell Signaling Technology, Danvers, MA, USA), anti-p16^Ink4a^ (1:500, Santa Cruz Biotechnology, Paso Robles, CA, USA), anti-PGC–1α (1:1000, Invitrogen, Carlsbad, CA, USA), anti-PPARγ (1:1000, Cell Signaling Technology, MA, USA), anti-H3 (1:1000, Abcam, M, USA), anti-ATG−7 (1:1000, Cell Signaling Technology, MA, USA), anti-ATG−12 (1:1000, Cell Signaling Technology, MA, USA), anti-p−H3 (1:1000, Cell Signaling Technology, MA, USA), anti-Cyclin D2 (1:2000, ABclonal Technology, Wuhan, China), anti-LC3 (1:500, Novus Biologicals, Littleton, CO, USA), anti-HA (1:5000, Proteintech, Wuhan, China), anti-FLAG (1:1000, Sigma-Aldrich, St. Louis, MO, USA), and anti-GAPDH (1:5000, Abclonal, Wuhan, China).

### 4.4. SA-β-Gal Staining

Senescence-associated galactosidase activity was measured as described previously [41]. Briefly, cultured cells were washed in 1× PBS and fixed for 3–5 min (room temperature) in 2% formaldehyde and 0.2% glutaraldehyde. Fixed cells were stained with fresh stain solution for SA-β-galactosidase activity at 37 °C for 4 h. The percentage of cells positive for SA-β-Gal staining was quantified and statistically analyzed (n = 3).

### 4.5. Oil Red O Staining

The cells were fixed at room temperature using 4% paraformaldehyde for 5 min, placed in an Oil Red O (1320-06-5, Sigma-Aldrich, MO, USA) solution (0.5% *w*/*v*, diluted with ddH_2_O at a ratio of 3:2) for 5 min at 25 °C, and then washed with 60% isopropanol. After staining the nuclei with hematoxylin for 30 s at 25 °C, the cells were washed with distilled water until the nuclei appeared blue. The morphological characteristics of the cells were observed under an optical microscope (Nikon Corporation, Tokyo, Japan). Lipid accumulation was quantitated after isopropanol extraction of Oil Red O from stained cells and optical density determinations made at 490 nm.

### 4.6. Disruption of the PGC–1α Gene through CRISPR/Cas9

Mouse PGC–1α sgRNA#1 5′-CTCAGCTACAATGAATGCAG (PAM)-3′, sgRNA#2 5′-GGAAGGGTTCTTACTAGAGA (PAM)-3′, sgRNA#3 5′-GAATGAGGCAAACTTGCTAG (PAM)-3′, and sgRNA#4 5′-CTTACCTCAAATATGTTCGC (PAM)-3′ were designed online (https://chopchop.rc.fas.harvard.edu/ accessed on 24 November 2022) and cloned into lentivirus CAS9 vector LentiCRISPRv2puro (#89290, Addgene, Watertown, MA, USA). The vectors were delivered into late-stage *p53^S/+^*+HRas cells by infection with lentivirus produced in 293T cells. Infected cells were selected by puromycin for several days and single clones were selected and amplified. The targeted clones were screened by limited dilution, and the disruption of gene expression was confirmed by sequencing of the targeting loci:PCR primers #1: 5′-TTTCTTGCTTTCCCTTTTTCTG-3′ and 5′-ACCCCTATCCTCCCCACTAATA-3′,PCR primers #2: 5′-TTGATGCACTGACAGATGGAG-3′ and 5′-ACAGAATGGGCAAATCTAGGAA-3′,PCR primers #3: 5′-TCAACCCACTCATGTCTTCTGT-3′ and 5′-TACTAGAGACGGCTCTTCTGCC-3′,PCR primers #4: 5′-CCAGATCTTCCTGAACTTGACC-3′ and 5′-CTCCCCATACATCAGTCAGACA-3′.

### 4.7. Immunofluorescence and Microscopy Analysis

Cells were spread onto coverslips and treated with or without DFO for different times. After washing twice in PBS, the cells were fixed with 3% paraformaldehyde–2% sucrose for 10 min at room temperature and permeabilized with 1% NP40 in PBS for 5 min at room temperature. After blocking in 5% BSA, the coverslips were incubated with an anti-p53 antibody (1:500, 9282, Cell Signaling Technology, MA, USA) overnight at 4 °C in a humid chamber and then with Alexa Fluor 568-conjugated anti-rabbit IgG secondary antibody (Life technologies corporation, Frederick, MD, USA) for 1 h at room temperature in the dark. Slides were mounted in VECTASHIELD mounting medium (H-1200, Vector Labs, Newark, CA, USA). Images were captured on a Nikon Ti-E microscope using equal exposure times for all images.

### 4.8. Real-Time PCR

For genotyping and copy number variation, genomic DNA was extracted from *p53^S/+^*+HRas or *p53^S/S^*+HRas cells. RT-PCR primers were designed at the *loxP* site, which specifically amplified the *p53S* allele. Forward: 5′-GGAACTTCCCGCGGATAAC-3′ and reverse: 5′-AATCAGTTTATCCTCCCTTTCACC-3′.

Total RNA was isolated from cells with TRIzol reagent (Invitrogen, CA, USA), and 1 μg of RNA was used for cDNA synthesis using a GoScript kit (Promega, Madison, WI, USA). cDNA was amplified using the following PCR primers:*PGC–1α* forward: 5′-AGCCGTGACCACTGACAACGAG-3′, reverse: 5′-GCTGCATGGTTCTGAGTGCTAAG-3′;*Sod1 forward*: 5′-CAAGCGGTGAACCAGTTGTG-3′, reverse: 5′-TGAGGTCCTGCACTGGTAC-3′;*Sod2 forward*: 5′-GCCTGCACTGAAGTTCAATG-3′, reverse: 5′-ATCTGTAAGCGACCTTGCTC-3′;*Ucp2 forward*: 5′-CAGGTCACTGTGCCCTTACCAT, reverse: 5′-CACTACGTTCCAGGATCCCAAG-3′;*Ant forward*: 5′-TTCCTGGCAGGTGGCATCG-3′, reverse: 5′-GGATTCTCACGACACAATCAATG-3′; and*Catalase forward*: 5′-ACCCTCTTATACCAGTTGGC-3′, reverse: 5′-GCATGCACATGGGGCCATCA-3′.

mRNA levels were standardized to those of GAPDH, and the data were analyzed based on the 2^−ΔΔCT^ values.

### 4.9. Chromatin Immunoprecipitation

Briefly, chromatin from cells was cross-linked. Precleared input proteins were immunoprecipitated with p53 (#OP03, Calbiochem, MA, USA). Complexes were recovered with beads (protein A agarose/salmon sperm DNA, #16-157, Merck Millipore, MA, USA), which were then washed. Cross-linking was reversed and the cleared DNA was recovered via standard precipitation approaches. The PGC–1α promoter was amplified with the following PGC1α-CHIP primers: forward: 5′-AGCCAGTCTCAATTAGACCTCA-3′ and reverse: 5′-AATTTGAACATCAGACACAATCAGA-3′ by SYBR green Q-PCR.

### 4.10. ROS Detection

Cellular ROS levels were measured by flow cytometry using DCFH-DA, a fluorescent probe. After washing twice in PBS, the cells were incubated with 10 μM DCFH-DA for 30 min at 37 °C, washed, resuspended in PBS, and then analyzed using a BD Accuri C6 Plus flow cytometer (BD Biosciences, San Jose, CA, USA).

### 4.11. JC-10

A JC-10 mitochondrial membrane potential assay kit (Sangon Biotech, Shanghai, China) was used to measure the mitochondrial membrane potential change. Briefly, cells were harvested and resuspended in 500 μL of JC-10 dye loading solution (with 2.5 μL 200×JC-10) and incubated at room temperature for 60 min in the dark. The fluorescence intensity was monitored by flow cytometry in the FL1 channel for green fluorescent monomeric signal detection (apoptotic cells), and in the FL2 channel for orange fluorescent aggregated signal detection (healthy cells). Cytometric data were analyzed with FlowJo V10.6.2 software.

### 4.12. ATP Detection Assay

The ATP content was measured using a Luminescent ATP detection kit (Beyotime, Shanghai, China) following the manufacturer’s instructions. Briefly, cells were lysed in lysis buffer and centrifuged at 12,000× *g* for 5 min at 4 °C. The ATP level in the supernatant was measured with a BD Accuri C6 Plus flow cytometer (BD Biosciences, CA, USA).

### 4.13. MitoTracker^TM^ Green FM Staining

Mitochondria were labeled with MitoTracker Green™ dye (#M7514, Thermo Fisher Scientific, MA, USA), which fluoresces after it accumulates in mitochondrial membrane lipids. Cells were incubated for 1 h at 37 °C with medium containing 200 nM MitoTracker Green. Following extensive washing with PBS, the cells were harvested, and the fluorescence was analyzed with a BD Accuri C6 Plus flow cytometer (BD Biosciences, CA, USA).

### 4.14. LysoTracker^TM^ Red DND-99 Staining

For lysosome staining, LysoTracker Red (#L5728, Thermo Fisher Scientific, MA, USA) was used according to the manufacturer’s instructions. Briefly, cells were incubated for 30 min at 37 °C with 50 nM LysoTracker^TM^ Red in Opti-MEM. Following extensive washing with PBS, the cells were harvested, and the fluorescence was analyzed with a BD Accuri C6 Plus flow cytometry (BD Biosciences, CA, USA).

### 4.15. Co-IP Experiments

To examine whether the interaction of PGC–1α and PPARγ was regulated by p53S in vitro, 293T cells were cotransfected with WT or mutant p53 and HA-PGC–1α and FLAG-PPARγ, and cell lysates were then incubated in TEB150 buffer (50 mM HEPES pH 7.3, 150 mM NaCl, 2 mM MgCl_2_, 5 mM EGTA, 0.5% Triton X-100, 10% glycerol, and proteinase inhibitors) for 6 h at 4 °C with anti-Tag or anti-Flag antibody cross-linked to agarose beads. After washing with TEB150 buffer three times, the bead eluate was analyzed by immunoblotting.

### 4.16. Statistical Analyses

The data are presented as the means ± standard deviations. The significance of differences was tested by one-way analysis of variance with Duncan’s multiple range test. The statistical significance threshold was a *p* value < 0.05. The data are shown as the mean ± standard deviation (SD).

## Figures and Tables

**Figure 1 ijms-24-03790-f001:**
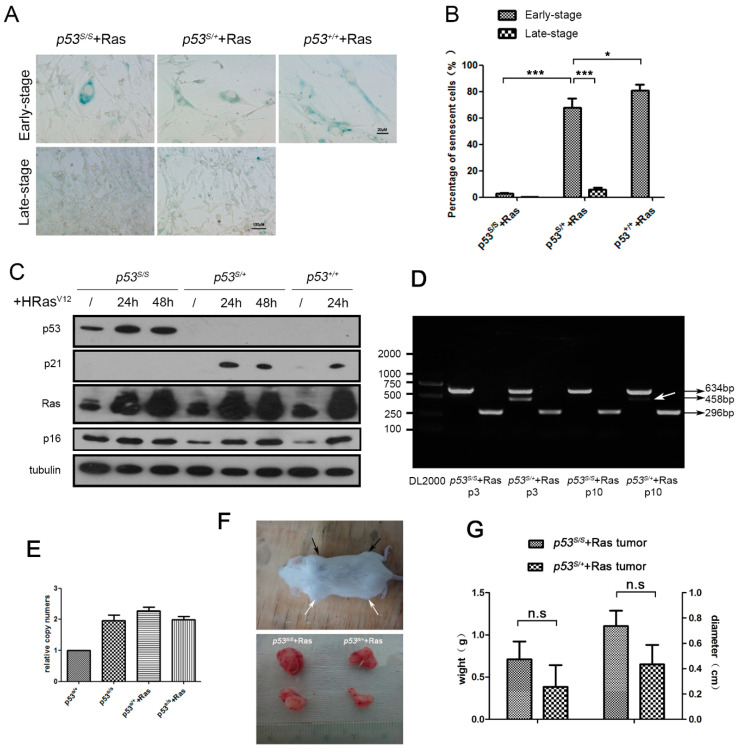
*p53^S/+^* cells escape from OIS and develop LOH. (**A**): SA-β-Gal staining of *p53^S/S^*, *p53^S/+^*, and *p53^+/+^* cells that had been infected with retrovirus transiently expressing HRas^V12^ (early-stage, passaging less than 15 days after infection) and stably (late-stage, passaging more than 30 days after infection). (**B**): The SA-β-Gal-positive cells were quantified and statistically analyzed. (**C**): Western blot analysis of *p53^S/S^*, *p53^S/+^*, and *p53^+/+^* cells transfected with HRas^V12^ for 24 h and 48 h. (**D**): Genotyping of *p53^S/+^*+Ras and *p53^S/S^*+Ras early-(p3) and late-(p10) stage cells. Restriction fragments of 458 bp are specific PCR products generated from the WT *p53* allele, and both 634 bp and 296 bp products of the p53S allele were obtained by PCR. LOH is indicated by a white arrow. (**E**): The *p53S* gene copy numbers were detected by real-time PCR. (**F**): In vivo tumorigenesis was established by the subcutaneous injection of cells into SCID mice. Upper panel: *p53^S/+^*+Ras (black arrows) and *p53^S/S^*+Ras (white arrows) cells formed fast-growing tumors within 1–2 weeks. Lower panel: *p53^S/+^*+Ras and *p53^S/S^*+Ras tumors dissected from a SCID mouse. (**G**): The weight and diameter of the *p53^S/+^*+Ras or *p53^S/S^*+Ras tumors from SCID mice were quantified and statistically analyzed. Values represent the means ± SDs of at least three independent experiments. Statistical significance was evaluated by Student’s *t*-test. * *p* < 0.05, *** *p* < 0.001, n.s.: not significant.

**Figure 2 ijms-24-03790-f002:**
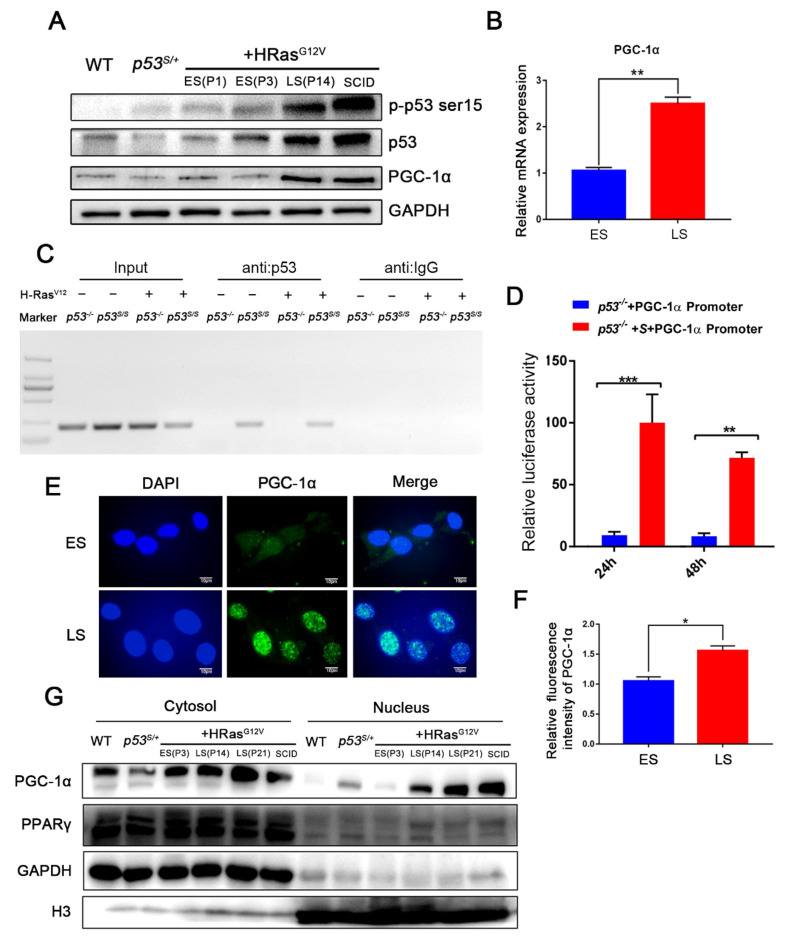
PGC–1α expression was upregulated in late-stage *p53^S/+^*+Ras cells. (**A**): Western blot analysis of protein expression in the indicated cell lines. ES: early-stage *p53^S/+^*+Ras cells, LS: late-stage *p53^S/+^*+Ras cells, SCID: a cell line derived from late-stage *p53^S/+^*+Ras cells that formed SCID tumors. (**B**): Real-time PCR analysis showed the expression level of PGC–1α. (**C**): A ChIP assay was used to determine the ability of p53S to bind to the promoter. (**D**): The promoter region of PGC–1α was cloned into a pEZX-GA01 reporter vector, and luciferase activity was measured to determine the effect of p53S on PGC–1α transcription. (**E**): Immunostaining for PGC–1α in ES and LS cells. (**F**): Quantitation of PGC–1α levels shown in (**E**). (**G**): The levels of PGC–1α and PPARγ in the nucleus and cytosol in ES and LS *p53^S/+^*+Ras cells. Values represent the means ± SDs of at least three independent experiments. The PPARγ antibody can be used for WB detection of endogenous PPARγ1(53KD) and PPARγ2(57KD). Statistical significance was evaluated by Student’s *t*-test. * *p* < 0.05, ** *p* < 0.01, *** *p* < 0.001.

**Figure 3 ijms-24-03790-f003:**
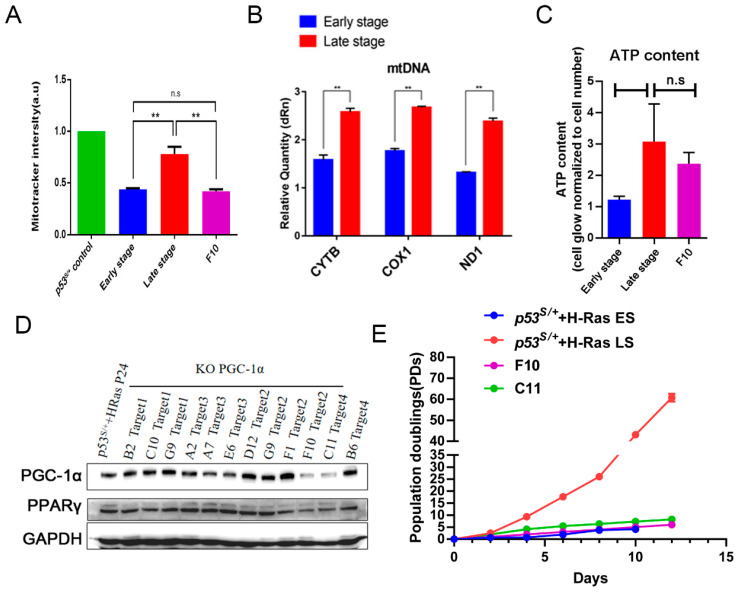
The quality and quantity of mitochondria in late-stage *p53^S/+^*+Ras cells was enhanced by p53S-induced PGC–1α production. (**A**): MitoTracker™ Green staining was used to determine the number of mitochondria in the indicated cells. (**B**): Real-time PCR measurement of the expression level of mt-DNA (*CYTB*, *COX1*, *ND1*) in ES and LS cells. (**C**): The ATP content was measured in ES, LS, and F10 (LS cells with *PGC–1α* gene knockdown) cells. (**D**): Western blot analysis of the expression of PGC–1α and PPARγ in PGC–1α-knockdown LS cell lines after colony screening, and the expression of two edited PGC–1α genes, “F10 Target2” and “C11 Target4”, was reduced. (**E**): NIH 3T3 assays were used to measure the proliferative capacities of the indicated cell lines. Values represent the means ± SDs of at least three independent experiments. Statistical significance was evaluated by Student’s *t*-test. ** *p* < 0.01, n.s.: not significant.

**Figure 4 ijms-24-03790-f004:**
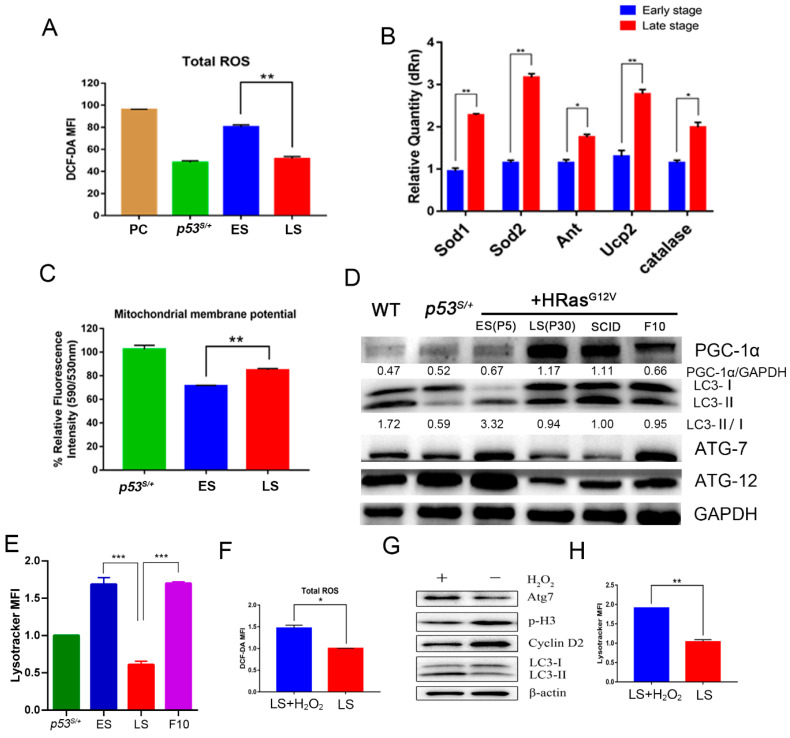
ROS induce autophagy in *p53^S/+^*+Ras cells. (**A**): Detection of ROS levels in ES and LS cells by DCFH-DA staining. PC: Positive control, *p53^S/+^* cells treated with H_2_O_2_. (**B**): Real-time PCR measurement of the expression levels of antioxidant enzymes (*Sod1*, *Sod2*, *Ant*, *Ucp2*, and *Catalase*) in ES and LS cells. (**C**): JC-10 staining was performed to detect the mitochondrial membrane potential of ES and LS cells. (**D**): Western blot analysis of the abundance of autophagy-related proteins in the indicated cell lines. (**E**): LysoTracker™ Red staining was performed to count the lysosomes in ES, LS, and F10 cells. (**F**): DCFH-DA staining was used to detect ROS in LS cells with or without H_2_O_2_ treatment. (**G**): Western blot analysis showing the levels of autophagy- and cell cycle-related proteins in LS cells with or without H_2_O_2_ treatment. (**H**): LysoTracker™ Red staining was performed to count the lysosomes in the LS cells with or without H_2_O_2_ treatment. Values represent the means ± SDs of at least three independent experiments. Statistical significance was evaluated by Student’s *t*-test. * *p* < 0.05, ** *p* < 0.01, *** *p* < 0.001.

**Figure 5 ijms-24-03790-f005:**
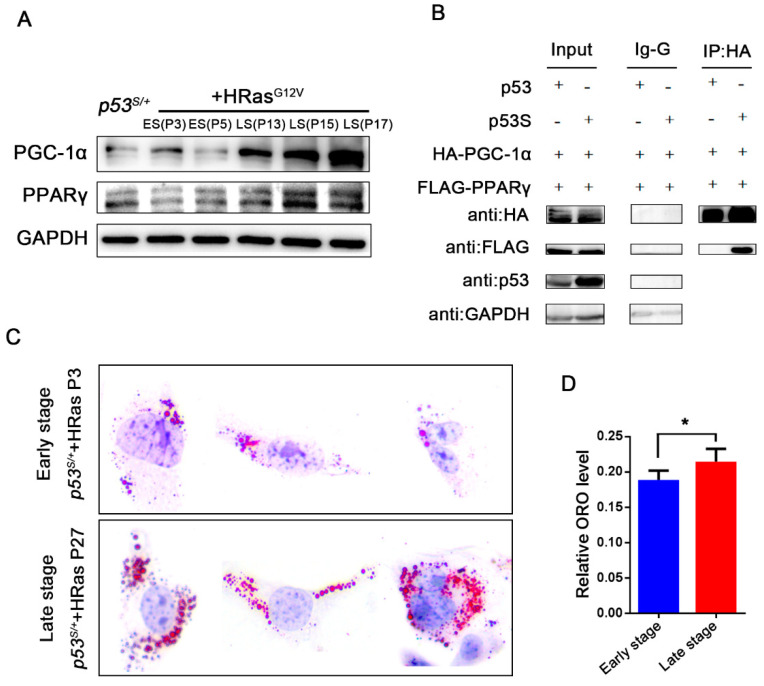
p53S enhances the interaction between PGC–1α and PPARγ. (**A**): Western blot analysis of the expression of PGC–1α and PPARγ in ES and LS cells. (**B**): Co-IP was performed to determine the interaction between HA-PGC–1α and FLAG-PPARγ with WT p53 or p53S overexpression in 293T cells. (**C**): Intracellular lipid accumulation in LS cells was measured by ORO staining. (**D**): Quantitation of the lipids presented in C. Values represent the means ± SDs of at least three independent experiments. Statistical significance was evaluated by Student’s *t*-test. * *p* < 0.05.

**Figure 6 ijms-24-03790-f006:**
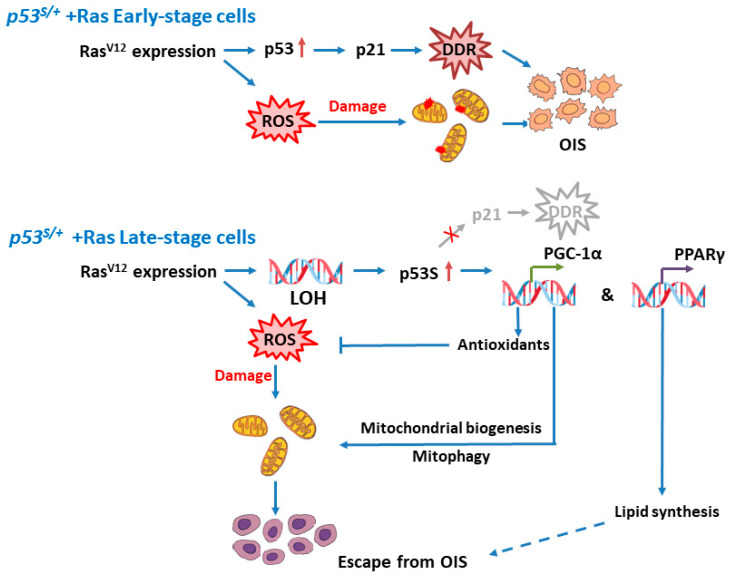
Schematic representation of how late-stage *p53^S/+^*+Ras cells escape OIS. In the initiation of Ras^V12^ expression, the wild-type p53 allele is activated and induces the DNA damage response (DDR) through its promotion of downstream gene expression. When the pressure of oncogene activation persists, *p53^S/+^* cells undergo a loss of heterozygosity (LOH), leading to the deletion of the p53 WT allele and high expression of p53S; however, the mechanism is unknown. Although p53S cannot mediate classical DDR, it can promote antioxidant gene expression by directly upregulating PGC–1α to reduce reactive oxygen species (ROS) levels, increase mitochondrial synthesis, downregulate autophagy level, and interact with PPARγ to regulate lipid synthesis. These processes provide favorable conditions for *p53^S/+^* cells to escape senescence and thus become tumorigenic. OIS: oncogene-induced senescence.

## Data Availability

Not applicable.

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
