# Peer review of "Gain-of-Function p53N236S Mutation Drives the Bypassing of HRasV12-Induced Cellular Senescence via PGC–1α"

_ijms, 2023, doi:10.3390/ijms24043790_

Round 1
Reviewer 1 Report
The authors found that p53N236S heterozygous mouse embryonic fibroblasts escaped HRasV12-induced senescence in vitro and formed tumors with subcutaneous injection into SCID mice. The authors further demonstrated p53S increased the level and nuclear translocation of PGC-1α in late-stage p53S/++ Ras cells, which promoting the biosynthesis and function of mitochondria by inhibiting senescence-associated ROS. The authors also suggested p53S regulated the interaction between PGC-1α and PPARγ to promote lipid synthesis. The observation and the PGC upregulation are convincing, yet the data to support PPAR-PGC interaction still need to be more solid. Also, it will be interesting to compare p53+/- vs p53S/+ MEFs to understand whether the finding is N246S allele specific or just loss of p53 wild type function. There are so minor issues as well.
1. 1. Figure 1B, p53S/S MEFs also showed low percentage of senescence, is it due to residual activity of p53S allele? Figure 1C, have the authors to test any other p53 mutations such as R172H, R245W et al ?
Figure 1E, Does the PCR in this experiment specifically detect N246S allele since the p53S/+ show one copy, then p53S/+ Ras showed 2 copies, suggesting the duplication of the mutant allele, is it true?
Figure F,G did not show the different of tumor formation, it will be better to have a early passage cells injections control.
2. 2. Figure 2G, is the band showed in cytosol fraction non-specific? can the authors provide a band picture of PPAR western blot? Also it will be better to have p53+/- mefs in the experiments.
3. 3. Figure 3D and 3E are redundant
4. 4. Is it possible the cells are undergoing senescence and also autophagy is activated?
5. 5. Figure 5B, the quality of the pictures is poor, why IP HA has not PGC bands? IP-flag, the authors can use PPAR antibody to do western to get a better picture. STE and LTE should be explained in figure legend.
Author Response
- Figure 1B, p53S/S MEFs also showed low percentage of senescence, is it due to residual activity of p53S allele? Figure 1C, have the authors to test any other p53 mutations such as R172H, R245W et al?
Figure 1E, Does the PCR in this experiment specifically detect N246S allele since the p53S/+ show one copy, then p53S/+ Ras showed 2 copies, suggesting the duplication of the mutant allele, is it true?
Figure F,G did not show the different of tumor formation, it will be better to have a early passage cells injections control.
Response: Figure 1B, it is possible that in the early-stage of HRas expression, cellular senescence can be induced by residual activity of p53S allele. However, the senescent cells are gradually replaced by proliferating cells after several passages. We haven’t tested if any other p53 mutations could bypass HRas induced senescence due to lack of p53 mutant hyterozygous MEFs. We tested with p53+/- MEFs, however, p53+/- cells were not able to escape from HRas induced senescence after several days of culture.
Figure 1E, exactly, the RT-PCR primers were designed at the loxP site of p53S allele, which specifically detect the mutant allele. We assume that the LOH caused by Ras would result in the loss of the wild-type allele, however, there was a duplication of the mutant allele signal. The mechanism still needs further exploration.
Figure F and G, since the early-stage p53S/++ Ras cells are senescent and proliferate very slowly, it is difficult to collect the amount of cells required for subcutaneous tumor injection. Therefore, we only performed a comparison of the tumor formation capacity of late-stage cells, which further suggesting the genotype of p53S/+ change into p53S/S rather than p53S/- via LOH in late-stage cells.
- Figure 2G, is the band showed in cytosol fraction non-specific? can the authors provide a band picture of PPAR western blot? Also it will be better to have p53+/- mefs in the experiments.
Response: According to the antibody's instructions, this PPARγ antibody can be used for WB detection of endogenous PPARγ1(53KD) and PPARγ2(57KD). We have illustrated in the figure legend. We agree that p53+/- cells should be used as control in the experiments, however, we don’t have this genotype MEF in stock, we will repeat this experiment as soon as we have it.
- Figure 3D and 3E are redundant
Response: Figure 3D has been removed.
- Is it possible the cells are undergoing senescence and also autophagy is activated?
Response: Recent studies on cross-talk between autophagy and senescence in fibroblasts show contradictory context-dependent results. It was showed that the inhibition of autophagy delays the Ras induced senescence(1, 2).
- Young AR, Narita M, Ferreira M, Kirschner K, Sadaie M, Darot JF, Tavare S, Arakawa S, Shimizu S, Watt FM, Narita M. Autophagy mediates the mitotic senescence transition. Genes Dev. 2009;23(7):798-803. Epub 2009/03/13. doi: 10.1101/gad.519709. PubMed PMID: 19279323; PMCID: PMC2666340.
- Bernard M, Yang B, Migneault F, Turgeon J, Dieude M, Olivier MA, Cardin GB, El-Diwany M, Underwood K, Rodier F, Hebert MJ. Autophagy drives fibroblast senescence through MTORC2 regulation. Autophagy. 2020;16(11):2004-16. doi: 10.1080/15548627.2020.1713640. PubMed PMID: WOS:000506961900001.
- Figure 5B, the quality of the pictures is poor, why IP HA has not PGC bands? IP-flag, the authors can use PPAR antibody to do western to get a better picture. STE and LTE should be explained in figure legend.
Response: Thanks for pointing out our deficiencies in Figure 5B. We did repeat experiments and got more reliable result to replace it.

Reviewer 2 Report
In this study the authors assess the role of the p53N236S mutation in bypassing oncogene-induced senescence. p53N236S is a known gain-of-function mutation that has been previously described and investigated regarding its effects on several parameters associated with cancer progression. The manuscript is generally well written and experiments have been adequately performed. I only have some concerns before the study can be considered for publication.
1. The manuscript contains some grammar and syntax errors that need to be corrected (for examples, please refer to the uploaded file).
2. I often miss some controls from presented experiments, i.e. p53+/+ cells or cells not infected with retrovirus transiently expressing HRasV12, depending on the case. I believe that the whole set of cells should be used in all assays (wt, p53S/+, p53S/S expressing or not HRas).
3. In order to elucidate the implication of Nrfs in the described phenomena, the authors should also perform experiments assessing the putative activation of the particular transcription factors, e.g. by their translocation into the nucleus. Their expression levels are not necessarily expected to alter.
4. The authors should update their references list and include some missing related to the work articles, e.g. FEBS Lett. 2018 Sep;592(18):3183-3197 and Genes (Basel). 2022 Apr 26;13(5):763.
Author Response
- The manuscript contains some grammar and syntax errors that need to be corrected (for examples, please refer to the uploaded file).
Response: Thank you very much for the comments, some grammar and spelling errors have been corrected.
- I often miss some controls from presented experiments, i.e. p53+/+ cells or cells not infected with retrovirus transiently expressing HRasV12, depending on the case. I believe that the whole set of cells should be used in all assays (wt, p53S/+, p53S/S expressing or not HRas).
Response: We agree that p53+/+ cells should be used as control in all assays, however, as we showed in Figure 1C, the cells quickly senesced and died because of H-Ras activated DDR, so we were unable to obtain cells after 24h of transfection for other experiments.
- In order to elucidate the implication of Nrfs in the described phenomena, the authors should also perform experiments assessing the putative activation of the particular transcription factors, e.g. by their translocation into the nucleus. Their expression levels are not necessarily expected to alter.
Response: We understand that further experiments should be performed to elucidate the implication of Nrfs in PGC-1α regulated OIS process, However, in the present study, we mainly focused on the mechanism of mitochondrial proliferation in late-stage p53S/++Ras cells. We think that PGC-1α downregulates autophagy levels in late-stage cells to reduce mitophagy may not be optimal, but should be sufficient to draw a conclusion that the activation of PGC-1α, leading to an increase in mitochondrial quality and quantity and promotion of p53S/++Ras cell bypass of OIS.
- The authors should update their references list and include some missing related to the work articles, e.g. FEBS Lett. 2018 Sep;592(18):3183-3197 and Genes (Basel). 2022 Apr 26;13(5):763.
Response: Thanks for the kindly reminder, we have updated the reference as No.15 and 40, and talked about the related work in the discussion.

Round 2
Reviewer 1 Report
The authors have answered my questions
Author Response
I am very grateful for all your comments on the manuscript.
Reviewer 2 Report
This is the revised version of a previously submitted work.
The authors have addressed most of my concerns raised during the previous round of the reviewing process.
Author Response

(The authors gave the same response as above.)
